# In Situ Imaging of Domain Structure Evolution in LaBGeO$_5$ Single Crystals

**Andrey Akhmatkhanov** [1] , **Constantine Plashinnov** [1]**, Maxim Nebogatikov** [1]**, Evgenii Milov** [2]**, Ilya Shnaidshtein** [2] **and Vladimir Shur** [1,*]

[1]  School of Natural Sciences and Mathematics, Ural Federal University, Ekaterinburg 620000, Russia; andrey.akhmatkhanov@urfu.ru (A.A.); steelzan@gmail.com (C.P.); maxneb@urfu.ru (M.N.)

[2]  Faculty of Physics, Lomonosov Moscow State University, Moscow 119991, Russia; milov@plms.phys.msu.su (E.M.); shnaidshtein@phys.msu.ru (I.S.)

*  Correspondence: vladimir.shur@urfu.ru; Tel.: +7-343-389-9568

**Abstract:** LaBGeO$_5$ (LBGO) crystals are unique ferroelectric materials for manufacturing highly efficient UV laser sources based on frequency conversion. This is due to their low cut-off wavelength, high nonlinear-optical coefficients, and non-hygroscopicity. Periodical poling requires a deep study of domain kinetics in these crystals. Domain imaging by Cherenkov second harmonic generation microscopy was used to reveal the main processes of domain structure evolution: (1) growth and merging of isolated domains, (2) growth of stripe domains formed on the artificial linear surface defects, and (3) domain shrinkage. In a low field, growth of triangular domains and fast shape recovery after merging were observed, while in a high field, the circular domains grew independently after merging. The revealed essential wall motion anisotropy decreased with the field. The anisotropy led to significant shape transformations during domain shrinkage in low field. The formation of short-lived triangular domains rotated by 180 degrees with respect to the growing isolated domains was observed. The obtained results were explained within the kinetic approach to domain structure evolution based on the analogy between the growth of crystals and ferroelectric domains, taking into account the gradual transition from determined nucleation in low field to the stochastic one in high field.

**Keywords:** ferroelectric; Cherenkov second harmonic generation; domain kinetics; domain shape; domain imaging; domain merging

## 1. Introduction

Borate-based nonlinear-optical crystals are the most promising candidates for the fabrication of efficient UV laser light sources based on frequency conversion (second harmonic generation and sum frequency generation). However, the applicability of the most popular borate-based crystals, such as β-BaB$_2$O$_4$ and CsLiB$_6$O$_{10}$, is limited due to their hygroscopicity and the light wave walk-off effect, which is inseparably linked to the angular phase matching utilized in these crystals. On the other hand, uniaxial ferroelectric LaBGeO$_5$ (LBGO) single crystals [1,2] are non-hygroscopic and allow realizing walk-off-free quasi-phase-matched frequency conversion, thus making it possible to use the highest available nonlinear optical coefficient: $d_{33}$ = 1.3 pm/V [3]. This type of conversion requires the creation of a precise periodical stripe ferroelectric domain structure (domain engineering) [4–6]. The early attempts at periodical poling in LBGO resulted in the creation of a domain pattern with a period of 5.5 μm in a 0.5-mm-thick sample for the first-order second harmonic generation (SHG) at 365 nm [7]. Periodical poling with a period of 2.1 μm for SHG at 266 nm was realized in a 0.3-mm-thick crystal with only 80% pattern homogeneity, which limited the output power and efficiency of the device [8,9].

Thus, deeper study of domain structure evolution in LBGO, including detailed imaging of domain kinetics, is required for further development of domain engineering methods in order to obtain highly homogeneous short-pitch patterns in thicker crystals.

LBGO single crystals undergo ferroelectric phase transition at 532 °C [10,11] and possess a stillwellite crystal structure with point symmetry $C_3$ [1]. Outstanding properties of LBGO crystals revealed after the first crystal growth in the 1990s [1,2] have stimulated strong scientific interest, resulting in detailed study of their thermal, dielectric [10], nonlinear-optical [3,12], structural [11,13], and ferroelectric properties [14,15].

The first polarization reversal in LBGO at room temperature was realized in Ref. [16]. The frequency and temperature dependences of the coercive field during polarization reversal were studied [16,17]. It was shown that the switching time followed the activation-type temperature dependence with an activation energy of 0.86 eV [17]. Polarization reversal in high field (>15 kV/mm) and at elevated temperatures was studied by analyzing the switching current data [18,19].

Optical microscopy after selective chemical etching was used for imaging of the static as-grown [19] and tailored periodical domain structures in LBGO [7,20]. Domain imaging using SHG interference microscopy with a spatial resolution of about 1 μm allowed revealing the artificially created periodic stripe domain structure with walls parallel to the X crystallographic direction [17] and studying its temperature stability [19]. The imaging of the domain structure with a high spatial resolution was only realized in one paper so far: Piezoresponse force microscopy has allowed demonstrating that polarization reversal in a field above 15 kV/mm leads to the formation of circular or irregularly shaped domains [18]. It should be noted that imaging of the domain structure evolution during polarization reversal has not been realized yet.

Cherenkov SHG microscopy is one of the most effective modern techniques used for ferroelectric domain imaging both at the surface and in the bulk. It is based on the registration of the SHG signal from the single domain wall by means of a confocal microscope [21]. It allows analyzing in detail the domain structure evolution and local tilt changes of charged domain walls with spatial resolutions down to 400 nm in various ferroelectric crystals [21–26].

In the present paper, we conducted the domain wall imaging by Cherenkov SHG microscopy for detailed study of the domain structure evolution in LBGO single crystals under the application of series of short electric field pulses.

## 2. Materials and Methods

The studied Z-cut 0.5-mm-thick plates of a single-domain LBGO single crystal grown by the Czochralski method (Oxide corp., Mukawa, Yamanashi, Japan) had dimensions of $5 \times 15$ mm². Liquid electrodes (LiCl saturated aqueous solution) were used for field application. An LBGO plate was glued over the 1-mm-in-diameter round hole in a supporting glass and sandwiched between two glass plates covered by transparent ITO electrodes [27] (Figure 1a). Artificial straight scratches at the sample surface were used as the nucleation centers for the creation of the stripe domains with various orientations.

Each polarization reversal has been realized by the application of series of identical short unipolar rectangular field pulses (Figure 1b) generated by analog-digital/digital-analog converter (ADC/DAC) board NI PCI 6251 (National Instruments, Austin, Texas, USA) and amplified by the high-voltage amplifier TREK 20/20C (Trek Inc., Lockport, New York, USA). Pulse rise/decrease time was about 30 ms. Pulse amplitude ($E_{ex}$) ranged from 3.75 to 16.25 kV/mm and pulse duration ($t_p$) from 0.12 to 15 s. The chosen pulse duration for the given amplitude allowed realizing complete switching of the area covered by an electrode by application of series of about 50 pulses. After each pulse, the domain structure (domain wall positions) was imaged by Cherenkov SHG microscopy (Figure 1b) (see Appendix A for experimental setup and image processing details). The area of $200 \times 200$ μm² was imaged with a spatial resolution of about 400 nm. The time interval between pulses in the series, equal to 9.5 s, was maintained for imaging of the static domain structure by Cherenkov SHG.

The restoration of the initial single-domain state was realized by the application of series of negative polarity pulses.

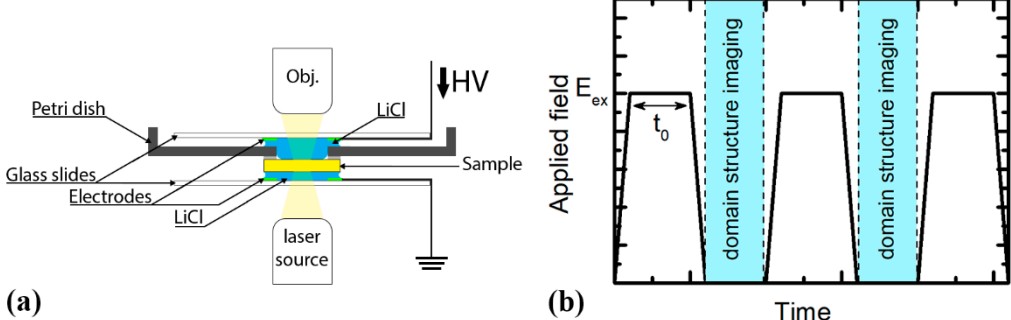

**(a)**　　　　　　　　　　　**(b)**

**Figure 1.** (**a**) The scheme of the experimental setup. (**b**) The scheme of the applied field pulse series with domain structure imaging intervals.

## 3. Results

A set of domain wall images obtained during the application of each pulse series was preprocessed for contrast enhancement (Figure 2) (see Appendix A). It should be noted that domain structure imaging in the bulk has demonstrated that the domain size does not change significantly with depth, indicating that the domain wall tilt with respect to the polar axis is below 0.1°. The "kinetic map" of the domain structure evolution during the switching process was obtained by overlapping the series of subsequent domain wall images (Figure 3) (see Video S1 in Supplementary Materials) [27]. The kinetic map contains domain wall positions at regular time intervals during the whole switching process. The local value of the sideways domain wall motion velocity is proportional to the distance between consecutive wall positions in the kinetic map.

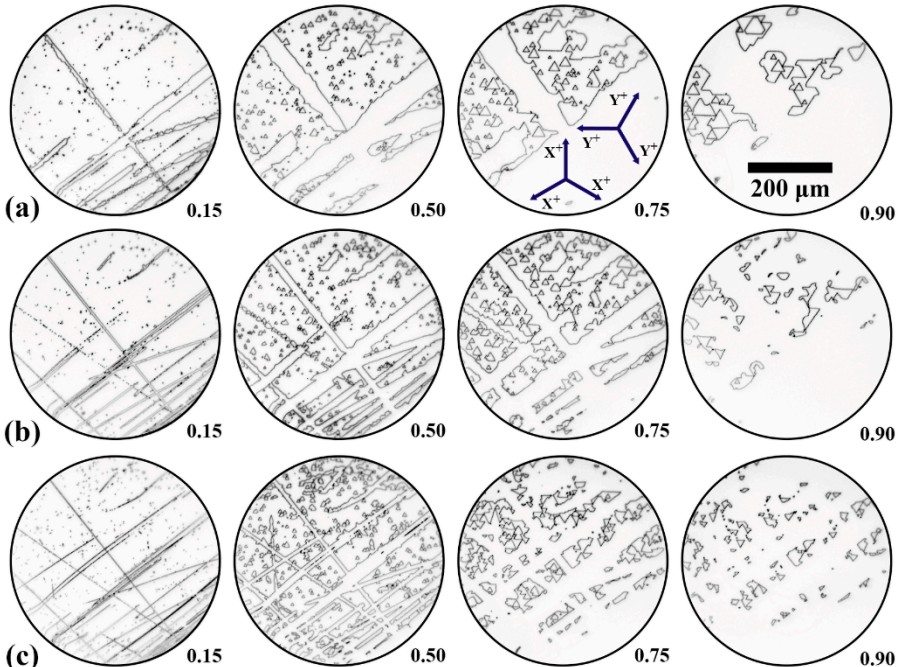

**Figure 2.** The domain wall images during the application of the pulse series with field amplitudes $E_{ex}$, kV/mm: (**a**) 5, (**b**) 7.5, and (**c**) 12.5. The fraction of the switched area for each image is indicated.

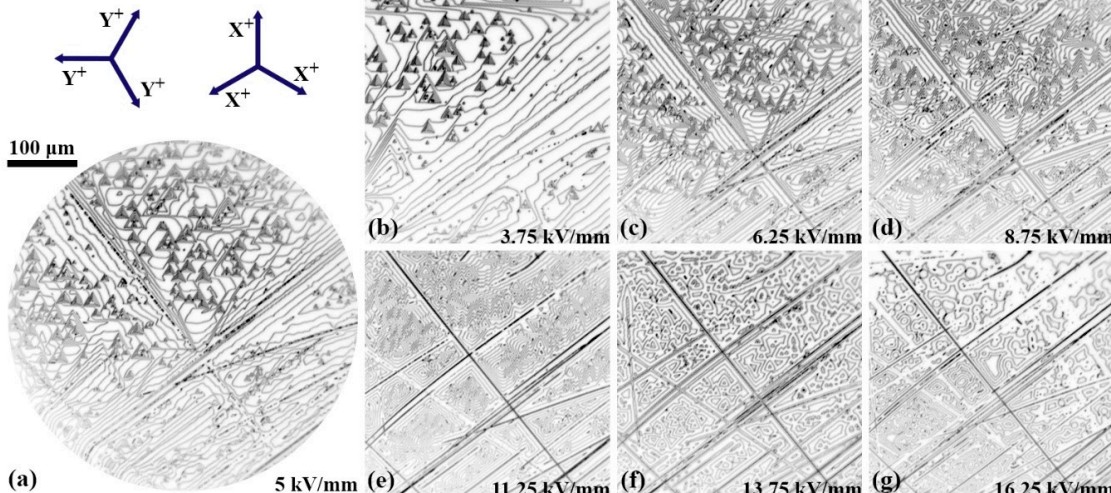

**Figure 3.** (**a**) The kinetic map of the polarization reversal process for the whole image area. The time interval between subsequent wall positions was 45 s. (**b**–**g**) Central fragments of the kinetic maps for various field values. Time intervals between the subsequent wall positions, s: (**b**) 160, (**c**) 15, (**d**) 5, (**e**) 1.75, (**f**) 1.25, and (**g**) 0.12.

Analysis of the kinetic maps (Figure 3) (see the complete image stacks in Video S2–S12 in Supplementary Materials) has allowed revealing the main processes of the domain structure evolution: (1) nucleation, growth, and merging of isolated domains, (2) formation and growth of stripe domains appearing along the linear surface defects, and (3) domain shrinkage.

### 3.1. Nucleation, Growth, and Merging of Isolated Domains

The appearance of the large number of isolated domains and their growth were observed over the whole switched area (Figure 3). The domain shape changed with the field from a regular triangle (Figure 4a,c) to a rounded triangle and, finally, to a circle (Figure 4b,c). The walls of regular triangles (designated as "slow Y− walls") are strictly oriented along Y crystallographic directions and move in X− directions. The similar transformation of the domain shape in high field from a polygonal one to a circular one has been obtained in uniaxial ferroelectrics with trigonal symmetry: Lead germanate $Pb_5Ge_3O_{11}$ [28] and congruent lithium tantalate [29].

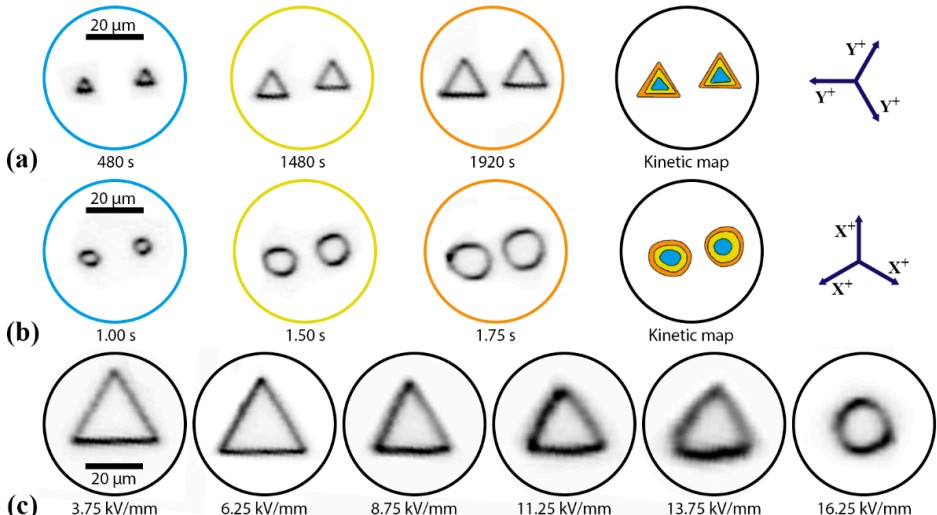

**Figure 4.** The domain wall images and kinetic maps for the growth of an isolated domain for $E_{ex}$ kV/mm: (**a**) 3.75 and (**b**) 16.25. (**c**) Field dependence of the shape of the isolated domain.

Analysis of the kinetic maps allowed measuring the field dependence of the slow Y− wall motion velocity (Figure 5). The obtained velocity range was from 2.7 nm/s for 3.75 kV/mm to 6.7 µm/s for 15 kV/mm (above three orders of magnitude).

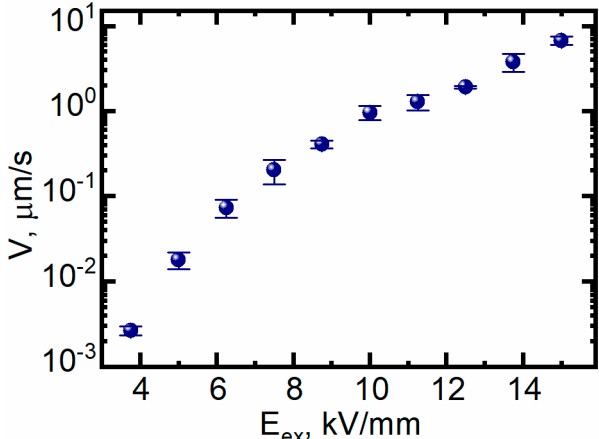

**Figure 5.** Field dependence of the slow Y− wall motion velocity of the isolated domain.

The scenario of isolated domain merging differs qualitatively for slow switching in low field and fast switching in high field.

In a low field (below 10 kV/mm), the domain "shape stability effect" is observed [29,30]. It represents the restoration of the triangular domain shape after domain merging due to the appearance and motion of the fast domain walls (designated as "fast Y+ walls") strictly oriented along Y crystallographic directions and moving in X+ directions. The fast Y+ walls are up to 40 times faster (for $E_{ex}$ = 3.75 kV/mm) than the slow Y− walls. As a result, the concave polygon appearing after merging transforms rapidly to the larger regular triangle (Figure 6a). For ferroelectrics of trigonal symmetry, such a merging scenario was obtained for hexagonal domains in lithium niobate [30–32] and for triangular domains in congruent lithium tantalate [33–35].

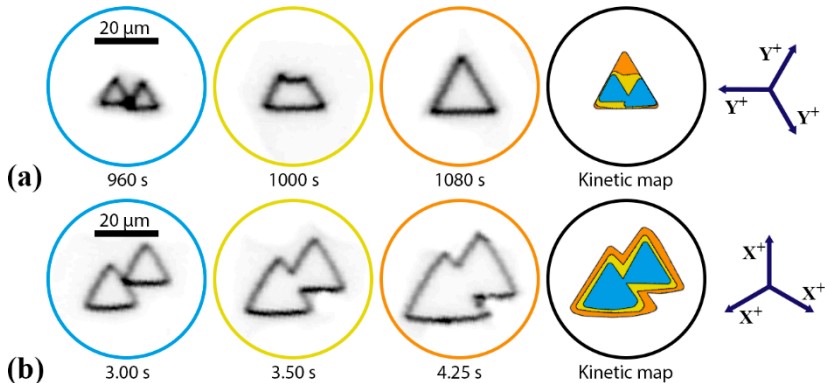

**Figure 6.** The domain wall images and kinetic maps for domain merging: (**a**) in low field, demonstrating the shape stability effect, $E_{ex}$ = 3.75 kV/mm, and (**b**) in high field, demonstrating the independent domain growth after merging, $E_{ex}$ = 12.50 kV/mm.

In a high field (above 10 kV/mm), the independent domain growth after merging was obtained, leading to the formation of the serrated domain walls (Figure 6b). The "formation of the serrated domain front" was observed previously in congruent lithium tantalate [33].

The ratio of velocities of fast Y+ domain walls to slow Y− walls decreases with the applied field (Figure 7). Thus, the domain merging in high field does not result in significant acceleration of domain

kinetics. This fact manifests itself in the smooth switching current without any oscillations or sharp peaks obtained in high field [18].

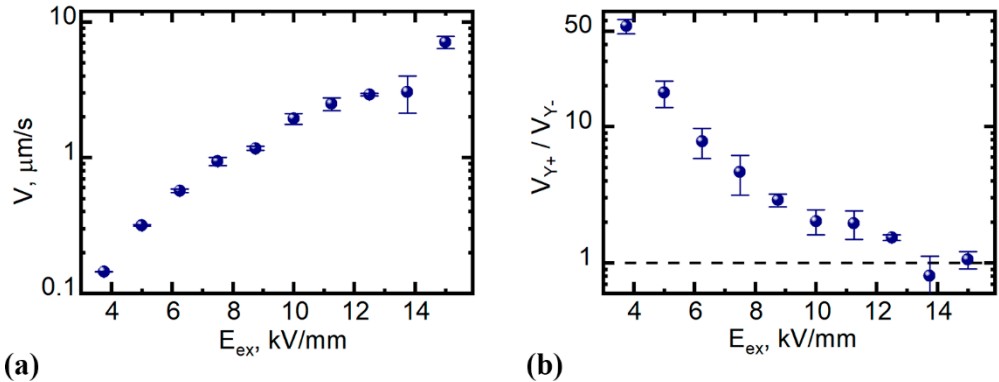

**(a)**　　　　　　　　　　　　　　　　　　**(b)**

**Figure 7.** Field dependences of (**a**) fast Y+ wall velocity and (**b**) ratio of fast Y+ to slow Y- wall velocities.

### 3.2. Formation and Growth of Stripe Domains

Artificial linear surface defects produced by mechanical treatment represent homogenously distributed nucleation sites, leading to an increase in the domain nucleation density by orders of magnitude. The fast merging of the domain chains that appeared allowed studying the growth of the stripe domains, which is important for periodical poling [7–9].

An essential anisotropy of the flat wall motion was obtained in low field for the stripe domains oriented close to the Y direction (Figure 8a,d). The fast walls (close to Y+ walls) were three times faster than the slow ones (close to Y− walls) for $E_{ex}$ = 5 kV/mm (Figure 8a,d). The wall motion anisotropy decreased with the field and disappeared for $E_{ex}$ > 15 kV/mm (Figure 8b–d). X-oriented flat walls did not demonstrate any motion anisotropy in the whole field range.

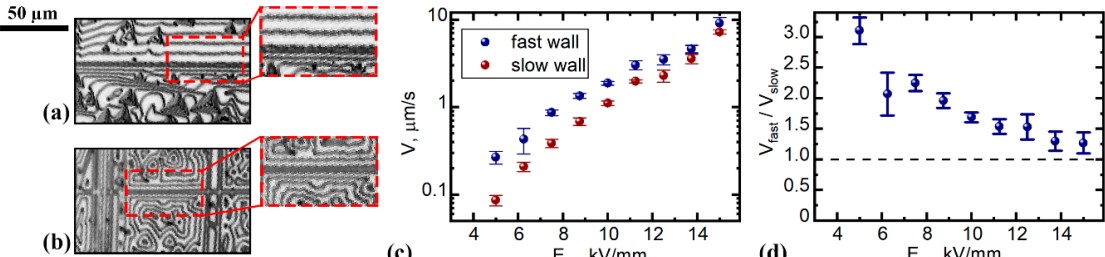

**Figure 8.** (**a**,**b**) Fragments of kinetic maps representing the growth of the stripe domains that appeared at the artificial linear surface defects, $E_{ex}$, kV/mm: (**a**) 6.25 and (**b**) 15. Time intervals between subsequent wall positions, s: (**a**) 15 and (**b**) 0.5. Field dependences of (**c**) fast (close to Y+) and slow (close to Y–) wall velocities, and (**d**) the ratio of fast to slow wall velocities.

### 3.3. Domain Shrinkage

The final stage of the domain structure evolution represents a shrinkage of the last domains with initial orientation of the spontaneous polarization. In low field, the polygonal domains with Y+ and Y− plane walls were formed (Figure 9a) and the anisotropy of wall motion velocity led to significant shape transformations (Figure 9a). Moreover, triangular domains with slow Y− walls rotated by 180 degrees with respect to the initial growing isolated domains were observed (Figure 9b).

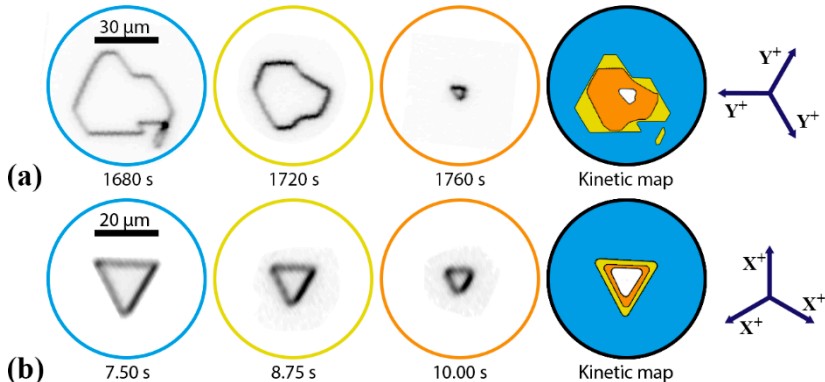

**Figure 9.** The domain shrinkage stage. The domain wall images and kinetic maps: (**a**) transformation of the polygonal domain shapes at $E_{ex} = 3.75$ kV/mm, and (**b**) evolution of the formed rotated triangular domain with slow Y− walls at $E_{ex} = 11.25$ kV/mm.

## 4. Discussion

The obtained results can be explained within the kinetic approach to domain structure evolution based on the analogy between the growth of crystals and ferroelectric domains [36–39]. The domain growth is governed by the nucleation process: 3D nucleation—formation of new domains; 2D nucleation at the domain wall—elemental step generation; and 1D nucleation—kink motion. Thus, the sideways wall motion is a result of step generation and kink propagation along the wall [39]. The faceted growth of polygon domains is a result of the determined nucleation, which represents the generation of the elemental steps at the domain vertices and kink motion in three directions [39]. The triangular domain shape in LBGO is determined by $C_3$ crystal symmetry. The similar domain shape is obtained in congruent lithium tantalate [29,33].

The field dependence of domain wall motion velocity for faceted domain growth in low field [39,40]:

$$V(E) = \mu(E_{ex} - E_{th}),\tag{1}$$

where $\mu$ is a domain wall mobility and $E_{th}$ is a threshold field.

The strong anisotropy of the wall mobility and the threshold field manifest themselves in the formation of superfast domain walls, which occurs (1) after the merging of isolated domains leading to the obtained shape stability effect [31], and (2) during shrinkage, resulting in significant shape transformations and the formation of the short-lived triangular domains with Y+ walls.

The obtained change of the domain shape in high field can be attributed to an increase in the input of the stochastic step generation mechanism with equal nucleation probability along the whole wall [29,41]. The domination of the stochastic nucleation leads to isotropic domain growth and the formation of the circular domains.

The field dependence of domain wall motion velocity for isotropic domain growth in high field [41]:

$$V(E) = V_{max} \exp\left(-\frac{E_{ac}}{E_{ex} + E_b}\right),\tag{2}$$

where $E_{ac}$ is an activation field, $V_{max}$ is a domain wall velocity in the limit of high field, and $E_b$ is an internal bias field.

The fitting of experimentally measured wall motion velocities in the whole field range allowed revealing two field regions. In low field regions ($E_{ex} < 12$ kV/mm), the experimental data were fitted by Equation (1) (Figure 10) with $\mu = (0.35 \pm 0.04) \cdot 10^{-3}$ mm$^2$/(kV·s) and $E_{th} = (7.4 \pm 1.2)$ kV/mm. In high field regions ($E_{ex} > 12$ kV/mm), the data were fitted by Equation (2) (Figure 10) with $V_{max} = (3.7 \pm 0.1)$ mm/s and $E_{ac} = (94 \pm 1)$ kV/mm. The $E_b$ parameter was fixed to 0 kV/mm during fitting.

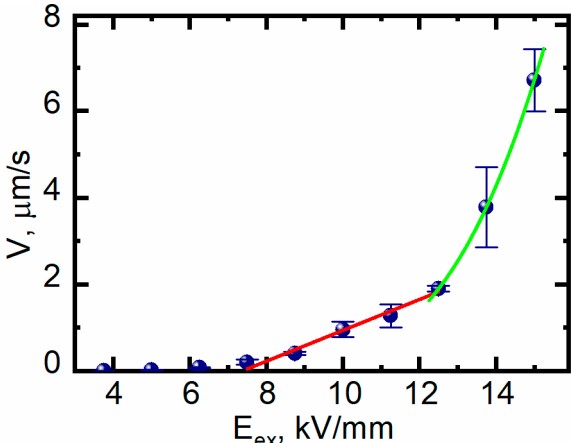

**Figure 10.** Field dependence of the slow Y- wall velocity fitted by Equation (**1**) (red line) and Equation (**2**) (green line).

The slight shift of the obtained low-high field boundary (12 kV/mm), as compared with that for the domain merging scenario (10 kV/mm), can be attributed to the higher sensitivity of the domain merging to the input of stochastic nucleation.

The extremely slow "subthreshold switching" for $E_{ex} < 7.4$ kV/mm is related to slow bulk screening of the depolarization field, which leads to the efficient decrease in the threshold field [42].

The change of the domain shape with the field ("barrel distortion") was characterized quantitatively (see Appendix B) (Figure 11). The obtained increase in the distortion can be attributed to an increase in the relative input of stochastic nucleation [29].

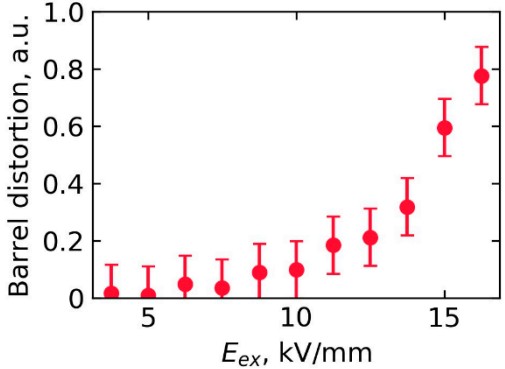

**Figure 11.** The field dependence of the barrel distortion of the isolated domain (see Appendix B).

## 5. Conclusions

We have carried out a detailed study of domain structure evolution in LBGO single crystals under the application of series of short electric field pulses using Cherenkov SHG microscopy for domain wall imaging. Analysis of the kinetic maps has allowed revealing the main processes of the domain structure evolution: (1) nucleation, growth, and merging of isolated domains, (2) formation and growth of stripe domains, and (3) domain shrinkage. It has been shown that the shape of individual domains changes with the field from regular triangular to rounded triangular and, finally, to circular. The walls of regular triangles ("slow Y− walls") were strictly oriented along Y crystallographic directions and moved in X− directions. The scenario of isolated domain merging differed qualitatively for switching in low and high fields. In low field, the domain restoration of the triangular domain shape after domain merging was observed. In high field, the independent domain growth after merging was obtained. It was shown that the shape restoration is due to the appearance and motion of the fast Y+ walls strictly oriented along Y directions and moving in X+ directions. The fast Y+ walls were up to 40 times faster

than the slow Y− walls. The ratio of the velocities of fast to slow walls decreased with the applied field. Artificial linear surface defects produced by mechanical treatment led to the formation and growth of the stripe domains, which is important for periodical poling. An essential anisotropy of the flat wall motion obtained in low field for the stripe domains oriented in Y direction decreased with the field. The final stage of domain structure evolution represented a shrinkage of the last domains with initial orientation of the spontaneous polarization. In low field, the polygonal domains with Y+ and Y− plane walls were formed, and the anisotropy of wall motion velocity led to shape transformations. The obtained results were explained within the kinetic approach based on the analogy between the growth of crystals and ferroelectric domains. The faceted growth of polygonal domains is a result of determined nucleation, representing the generation of the elemental steps at the domain vertices and kink motion in three directions. The obtained change of the domain shape in high field was attributed to an increase in the input of the stochastic equiprobable step generation mechanism. The fitting of experimentally measured slow wall motion velocities allowed revealing two field regions. In low field, the experimental data were fitted by linear dependence and, in high field, by activation type dependence. The extremely slow "subthreshold switching" was related to slow bulk screening of the depolarization field, leading to a decrease of the threshold field.

**Supplementary Materials:** The following are available online at http://www.mdpi.com/2073-4352/10/7/583/s1. Video S1: An example of the kinetic map construction; Video S2–S12: Stacks of domain wall images obtained during polarization reversal in the corresponding field with a given time interval between frames (field and time interval are written in the file names).

**Author Contributions:** Conceptualization, V.S. and A.A.; methodology, A.A.; investigation, C.P. and M.N.; resources, E.M. and I.S.; visualization, C.P.; formal analysis, C.P. and A.A.; writing—original draft preparation, A.A. and V.S.; writing—review and editing, A.A. and V.S.; supervision, V.S.; project administration, A.A. All authors have read and agreed to the published version of the manuscript.

**Funding:** This research was funded by the Russian Science Foundation, grant number 19-12-00210.

**Acknowledgments:** The equipment of the Ural Center for Shared Use "Modern Nanotechnology" was used.

**Conflicts of Interest:** The authors declare no conflict of interest. The funders had no role in the design of the study; in the collection, analyses, or interpretation of data; in the writing of the manuscript; or in the decision to publish the results.

**Appendix A. SHG Cherenkov Domain Structure Imaging**

Two-dimensional domain structure images were reconstructed by means of scanning Cherenkov SHG microscopy using a modified NTEGRA SPECTRA (NT-MDT, Russia) confocal microscope. We used a pulsed ytterbium fiber laser (ILMI 10P, IRE-Polus, Russia) with wavelength 1064 nm, delivering 2 ns pulses at a 5 MHz repetition rate, as an excitation source. Average pump power was 50 mW. The fundamental beam was focused by an objective lens (Olympus UPlanFL N $40 \times 0.75$ NA). The focus position was scanned in three dimensions: In the X-Y plane by a fast dual-axis galvanometer optical scanner and in the Z direction by translation of the objective lens that is mounted on a motorized stage. The size of the X-Y scan was $200 \times 200$ μm$^2$ with a spatial resolution of about 400 nm. The SHG signal was collected by a condenser lens with numerical aperture 0.62, and its intensity was measured by a photomultiplier. A color glass filter blocked the transmitted fundamental beam.

The resulting "raw" domain structure images represented the white domain walls over the dark background (Figure A1a). Further image processing included:

1. Color inversion for increasing the image clarity (Figure A1b).
2. Contrast enhancement.
3. Cropping to exclude the poorly focused outer regions of an image.
4. Rotation of the image to align the Y axis of the sample horizontally.

The resulting image is presented in Figure A1c.

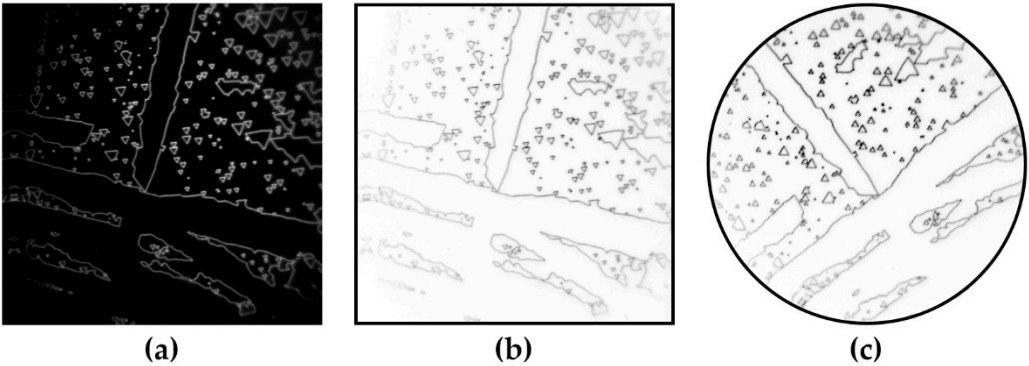

**Figure A1.** Domain wall images: (**a**) raw image obtained by Cherenkov SHG, (**b**) contrast inverted image, and (**c**) processed image.

## Appendix B. Calculation of the Barrel Distortion

The calculation of the barrel distortion was based on the representation of the shape of the isolated domain as a sum of a triangle and a circle. The following automatized procedure was used:

1. The isolated domain wall image (Figure A2a) was binarized using the Li thresholding [43,44] implemented in the scikit-image library [45] (version 0.15.0) (Figure A2b).
2. The binarized domain image was obtained by the "binary_fill_holes" function (SciPy library [46], version 1.3.1) (Figure A2c).
3. A domain center was found as an average radius vector of all black pixels on the binarized domain image (the red dot in Figure A2c), and the binarized domain wall image (Figure A2b) was replotted in polar coordinates with respect to the domain center (Figure A2d).
4. The obtained plot was fitted by the sum of triangle and circle representations in polar coordinates (Figure A2d):

$$r(\varphi) = \frac{R_t}{2\sin\left(\frac{5}{6}\pi - mod\left(\varphi - \varphi_0, \frac{2}{3}\pi\right)\right)} + R_c, \tag{A1}$$

where $R_t$ is a circumradius of a triangle, $R_c$ is a radius of a circle, $\varphi_0$ is a triangle rotation angle, and $mod\left(\varphi - \varphi_0, \frac{2}{3}\pi\right)$ is a remainder of an integer division of $(\varphi - \varphi_0)$ by $\frac{2}{3}\pi$.

5. The barrel distortion ($\gamma$) is calculated as:

$$\gamma = \frac{R_c}{R_t + R_c}. \tag{A2}$$

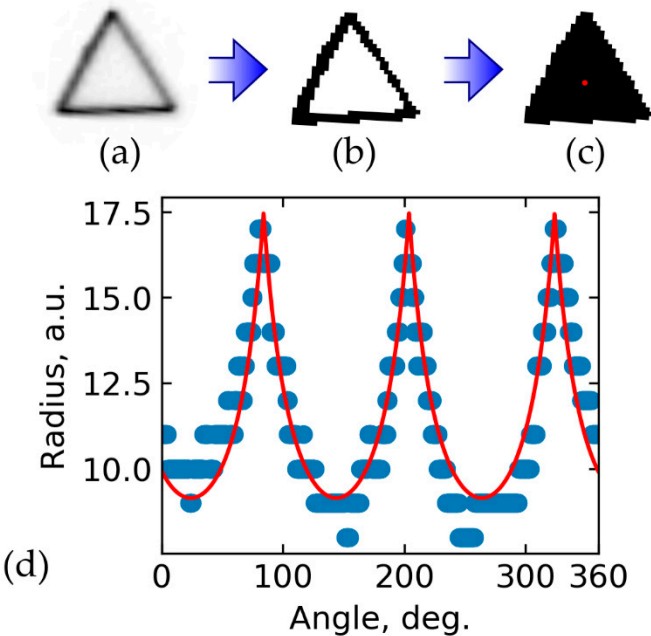

**Figure A2.** (**a**) An image of the domain walls of the isolated domain. (**b**) Binarized domain walls image. (**c**) Binarized domain image. (**d**) Representation of (**b**) in polar coordinates, solid red line—fitting by Equation (A1).

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
