# Peer review of "In Situ Imaging of Domain Structure Evolution in LaBGeO5 Single Crystals"

_crystals, doi:10.3390/cryst10070583_

Round 1
Reviewer 1 Report
The paper of Andrey Akhmatkhanov et al. reports the in situ imaging by means of Cherenkov second harmonic generation microscopy of domain evolution in LBGO ferroelectric crystals under electric fields of different intensity comparable to those used to obtain periodical poling in LBGO crystals.
The authors conclude that the evolution of the domains falls in two different regimes: one is a low field regime for which the domain wall velocity is linearly dependent on the applied field overcoming a threshold and the one at high fields is exponential .
Another interesting observation is the domain wall motion anisotropy at low field which decreases at higher electric fields. This anisotropy leads to significant shape transformations during domain shrinkage in low field.
The authors are able to reasonably fit the wall speed at the low and high field and suggest the domain evolution as a kind of crystal growth process.
Other information is reported in the two appendices and in the supplementing material reporting very nice videos of the domain evolution and of the obtained kinetic maps.
The paper, apart some not completely smooth English, is well written, contains a lot of new material and references and is very nice for the visual content.
Apart some English polishing the paper can be published in the present form.
Author Response
We would like to thank the reviewer for the interest to our work and for reasonable comments. We have made the minor revisions of the English along the manuscript text:
Page 1, line 36:
Before: “which allows utilizing”
After: “thus making it possible to use”
Page 1, line 38:
Before: “creation of”
After: “creating”
Page 1, line 41:
Before: “generation”
After: “SHG”
Page 1, line 45:
Before: “methods to obtain”
After: “methods in order to obtain”
Page 3, line 110:
Before: “appeared”
After: “appearing”
Page 5, line 138:
Before: “appeared”
After: “appearing”
Page 5, line 145:
Before: “In high field (above 10 kV/mm), the independent domain growth after merging leading to formation of the serrated domain walls is obtained (Fig. 6b).”
After: “In high field (above 10 kV/mm), the independent domain growth after merging is obtained leading to formation of the serrated domain walls”
Page 6, line 162:
Before: “The wall motion anisotropy was absent for X oriented flat walls in the whole field range.”
After: “X oriented flat walls did not demonstrate any motion anisotropy in the whole field range.”
Additionally, we have made some slight modifications of definite and indefinite articles throughout manuscript text in the "Track Changes" mode in Microsoft Word.
Reviewer 2 Report
The manuscript deals with analysis of domain structure evolution in Lanthanum Borogermanate Crystals. This compound being an object of interest of research since early nineties of the 20th Century is considered to be a perspective material for UV laser technology. Intense studies of properties of this compound along with rare earth ions doped systems have been carried out since that time. This work contains results on the imaging of domains by means of the Cherenkov second harmonic generation microscopy. Such features of domain structure in these crystals as growth and merging of isolated domains, growth of stripe domains formed on the artificial linear surface defects, and domain shrinkage were studied.
Manuscript contains new and interesting results that complement known properties of crystals of this material.
The manuscript can be accepted for publication.
Author Response
We are very happy about the positive feedback and that the reviewer considers our results interesting and supports publication.
Reviewer 3 Report
The manuscript by Akhmatkhanov et. al provides an interesting report on the imaging of domain evolution in LaBGeO5 crystals aided by in-situ imaging. The results are very clearly presented and appropriate mechanisms seem to have been applied and rationalised to justify the observed data. Results are of good quality (non-contentious) and interpretation in terms of existing models of domain growth are appropriate. Overall, I am happy with the experimental data as well as robust analysis presented in the manuscript. One minor suggestion is to add a conclusion section or concluding remarks at the end of the manuscript which would provide closure to the results and highlight the key achievements.
Otherwise, I recommend the publication of this manuscript in the present form.
Author Response
We’d like to thank the reviewer for the interest to our work and valuable comments. We have added the conclusion section (section â„– 4) to the end of the manuscript:
- Conclusion
We have carried out a detail study of domain structure evolution in LBGO single crystals under application of series of short electric field pulses using the Cherenkov SHG microscopy for domain wall imaging. Analysis of the kinetic maps has allowed revealing the main processes of the domain structure evolution: (1) nucleation, growth, and merging of isolated domains, (2) formation and growth of stripe domains, and (3) domain shrinkage. It has been shown that the shape of individual domains changes with field from regular triangular to rounded triangular and, finally, to circular. The walls of regular triangles (“slow Y- walls”) were strictly oriented along Y crystallographic directions and moved in X- directions. The scenario of isolated domain merging differed qualitatively for switching in low and high fields. In low field, the domain restoration of the triangular domain shape after domain merging was observed. In high field, the independent domain growth after merging was obtained. It was shown that the shape restoration is due to appearance and motion of the fast Y+ walls strictly oriented along Y directions and moving in X+ directions. The fast Y+ walls were up to 40 times faster than the slow Y- walls. The ratio of velocities of fast to slow walls decreased with applied field. Artificial linear surface defects produced by mechanical treatment led to formation and growth of the stripe domains, which is important for periodical poling. An essential anisotropy of the flat wall motion obtained in low field for the stripe domains oriented in Y direction decreased with field. The final stage of domain structure evolution represented a shrinkage of last domains with initial orientation of the spontaneous polarization. In low field, the polygonal domains with Y+ and Y- plane walls were formed, and the anisotropy of wall motion velocity led to shape transformations. The obtained results were explained within the kinetic approach based on the analogy between growth of crystals and ferroelectric domains. The faceted growth of polygonal domains is a result of determined nucleation representing the generation of the elemental steps at the domain vertices and kink motion in three directions. The obtained change of the domain shape in high field was attributed to increase in the input of the stochastic equiprobable step generation mechanism. The fitting of experimentally measured slow wall motion velocities allowed revealing two field regions. In low field, the experimental data were fitted by linear dependence, while in high field – by activation type dependence. The extremely slow “subthreshold switching” was related to slow bulk screening of depolarization field, leading to decrease of the threshold field.